# Innovative Strategies for Fertility Preservation in Female Cancer Survivors: New Hope from Artificial Ovary Construction and Stem Cell-Derived Neo-Folliculogenesis

**DOI:** 10.3390/healthcare11202748

**Published:** 2023-10-17

**Authors:** Stefano Canosa, Alberto Revelli, Gianluca Gennarelli, Gennaro Cormio, Vera Loizzi, Francesca Arezzo, Easter Anna Petracca, Andrea Roberto Carosso, Danilo Cimadomo, Laura Rienzi, Alberto Vaiarelli, Filippo Maria Ubaldi, Erica Silvestris

**Affiliations:** 1IVIRMA, Global Research Alliance, LIVET, 10126 Turin, Italy; aerre99@yahoo.com (A.R.); gianluca.gennarelli@unito.it (G.G.); 2Gynecology and Obstetrics 2U, Department of Surgical Sciences, S. Anna Hospital, University of Turin, 10126 Turin, Italy; 3Gynecology and Obstetrics 1U, Physiopathology of Reproduction and IVF Unit, Department of Surgical Sciences, S. Anna Hospital, University of Turin, 10126 Turin, Italy; acarosso@cittadellasalute.to.it; 4Gynecologic Oncology Unit, IRCCS Istituto Tumori “Giovanni Paolo II”, 70124 Bari, Italy; gennaro.cormio@uniba.it (G.C.); vera.loizzi@uniba.it (V.L.); easteranna97@hotmail.it (E.A.P.); e.silvestris@oncologico.bari.it (E.S.); 5Department of Interdisciplinary Medicine (DIM), University of Bari “Aldo Moro”, 70121 Bari, Italy; 6Obstetrics and Gynecology Unit, Department of Biomedical Sciences and Human Oncology, University of “Aldo Moro”, 70124 Bari, Italy; 7IVIRMA, Global Research Alliance, GENERA, Clinica Valle Giulia, 00197 Rome, Italy; cimadomo@generapma.it (D.C.); rienzi@generapma.it (L.R.); vaiarelli@generapma.it (A.V.); ubaldi@generapma.it (F.M.U.); 8Department of Biomolecular Sciences, University of Urbino “Carlo Bo”, 61029 Urbino, Italy

**Keywords:** cryopreservation, fertility preservation, oncofertility, cancer survivor, artificial ovary, stem cells

## Abstract

Recent advances in anticancer treatment have significantly improved the survival rate of young females; unfortunately, in about one third of cancer survivors the risk of ovarian insufficiency and infertility is still quite relevant. As the possibility of becoming a mother after recovery from a juvenile cancer is an important part of the quality of life, several procedures to preserve fertility have been developed: ovarian surgical transposition, induction of ovarian quiescence by gonadotropin-releasing hormone agonists (GnRH-a) treatment, and oocyte and/or ovarian cortical tissue cryopreservation. Ovarian tissue cryostorage and allografting is a valuable technique that applies even to prepubertal girls; however, some patients cannot benefit from it due to the high risk of reintroducing cancer cells during allograft in cases of ovary-metastasizing neoplasias, such as leukemias or NH lymphomas. Innovative techniques are now under investigation, as in the construction of an artificial ovary made of isolated follicles inserted into an artificial matrix scaffold, and the use of stem cells, including ovarian stem cells (OSCs), to obtain neo-folliculogenesis and the development of fertilizable oocytes from the exhausted ovarian tissue. This review synthesizes and discusses these innovative techniques, which potentially represent interesting strategies in oncofertility programs and a new hope for young female cancer survivors.

## 1. Oncofertility

Every year around the world, more than two million women are diagnosed with cancer, with breast cancer being the most common women’s cancer worldwide, followed by cervical cancer, ovarian cancer and uterine cancer. Epidemiological evidence clearly shows that women surviving cancer during childhood, adolescence or young adult life have a significantly lower probability of having children compared with healthy women of the same age range [1]. Some antineoplastic treatments (chemotherapy, radiotherapy, or surgery), in fact, may impair ovarian function in different ways, leading to an increased risk of irreversible infertility due to premature ovarian insufficiency (POI), defined as oligo/amenorrhea for ≥4 months and follicle-stimulating hormone (FSH) levels > 25 IU/L on two occasions, four weeks apart, before the age of 40 years. In some malignancies, such as breast cancer, leukemia, lymphomas, neuroblastoma, etc., oncological treatments performed to control the disease are quite harmful for the ovary, and increase the risk of POI [2,3]. The onset of POI after anticancer treatment depends not only on the type of therapy, but also on the patient’s age and the ovarian follicular reserve (OR) at the time of treatment [4,5]. The American Society of Clinical Oncology (ASCO) classified chemotherapy drugs in relation to their ovarian toxicity, assigning the highest risk (>80%) to adjuvant breast cancer poli-chemotherapy with CMF (Cyclophosphamide, Methotrexate, Fluorouracil), CEF (Cyclophosphamide, Epirubicin, Fluorouracil) and CAF (Cyclophosphamide, Adriamycin, Fluorouracil). Indeed, alkylating agents are quite toxic for the ovary, being able to induce progressive and bulk OR depletion, particularly when administered for six months to women ≥ 40 years; in these cases, the defense of OR by gonadotropin-releasing hormone analogues (GnRH-a) is only partially effective in limiting ovarian damage [6,7]. Similarly toxic for the ovary are total body irradiation (TBI) and radiotherapy including the ovaries in the irradiation field: patients submitted to TBI or pelvic irradiation at high dose have a higher than 90% risk of POI development [8]; even in this case, ovarian transposition aimed at avoiding the direct irradiation of the ovaries is often only partially effective [9]. Within the ovarian follicle, both oocyte and granulosa cells are vulnerable to the damage caused by chemotherapy. Each class of chemotherapeutic agents may have different mechanisms of action on cancer cells, with the end result being to affect the cell cycle. Histological studies on human ovaries have shown that chemotherapy can cause the loss of primordial follicles and ovarian atrophy; oocyte death due to induced apoptosis was identified as the main mechanism responsible for the loss of germ cells, but also injury to blood vessels and focal ovarian cortical fibrosis may contribute to generate the detrimental consequences of cytotoxic therapies [10]. Several fertility preservation options ranging from routine to experimental strategies are now available to counteract the treatment-related infertility risk. Both the American Society of Clinical Oncology (ASCO) and the European Society for Medical Oncology (ESMO) discuss the application of administration of GnRH analogues, ovarian transposition, surgical methods of fertility protection performed before anticancer treatment or oocyte in vitro maturation (IVM) and recommend the cryopreservation of oocyte and embryo as the mostly employed procedures able to guarantee motherhood in post-puberal female cancer survivors [11,12]. Although these cryopreservation procedures appear effective in terms of successful pregnancies in healthy women (nearly 45% for cryopreserved oocytes and 30% for embryos) [13], the preparative controlled ovarian stimulation is not suitable in cases of pre-puberal girls as well as in circumstances requiring urgent anticancer treatments as for hematologic malignancies, which cannot be retarded in relation to the time necessary to induce multifollicular growth. 

## 2. Ovarian Tissue Cryostorage

Ovarian tissue cryostorage is a valuable option for fertility preservation in case of women who urgently need chemotherapy, as in neoadjuvant protocols [14]. The major advantage of using cryopreserved cortex includes its feasibility independent from the menstrual cycle, also in prepubertal girls, and without the need of a precise OR evaluation, as the ovarian cortex always contains a relevant number of follicles. Ovarian biopsy is usually performed by laparoscopy taking approximately one third of both ovaries: a sharp surgical blade is used to gently remove the medulla and cutting the remaining ovarian cortex into pieces of approximately 5 × 5 × 1–2 mm that allow cryoprotective agents (CPAs) to quickly penetrate into the tissue and reduce the damage exerted by low temperature on of the follicle pool. Although slow freezing has been the conventional cryopreservation technique for years, extensive loss of follicles and damage to stromal cells have been reported [15]. To optimize the freezing strategies, vitrification has been proposed as an alternative procedure; this method is based on the induction of ultra-rapid cooling in the presence of high concentration of cryoprotectants, and is apparently functional in preventing cell injury by producing a glass-like amorphic state, as well as in maintaining stromal integrity similar to fresh tissue [16,17,18]. Furthermore, Diaz-Garcia et al., evaluated the live birth rate following oocyte vitrification with respect to OTC and autologous transplantation in oncological patients undergoing gonadotoxic treatments, and reported that despite slightly higher results achieved after oocyte freezing, ovarian tissue vitrification should be considered an alternative, good option when oocyte cryostorage is not feasible [19]. Interestingly, a recently published systematic review and meta-analysis has reported a significantly greater proportion of intact stromal cells in vitrified tissue versus slow-frozen tissue, with no significant differences with respect to the proportion of intact primordial follicles, extent of DNA fragmentation, or mean primordial follicle density [20]. Conversely, some reports suggested that vitrification may induce changes in mRNA expression and delayed growth of follicles in vitro in human ovarian tissues [21,22].

More recently, utilization of slush nitrogen has been proposed to further improve vitrification efficacy; morphology, ultrastructure and viability of both follicles and stromal cells are apparently better preserved in defrosted components, as compared with the same procedure using liquid nitrogen [23,24]. However, the worldwide practice is still predominantly the slow freezing of ovarian tissue and only four children have been born after vitrification.

Ovarian tissue transplantation (OTT) after thawing can be performed either at heterotopic sites outside the pelvis (subcutaneous tissue of the forearm or of the lower abdomen, sovrapubic subperitoneal space, or subfascial space of the abdominal rectus or of the pectoralis muscle), or at orthotopic sites (the remnant atrophic ovary, the broad ligament, the serosa of Fallopian tubes, or the pelvic wall at the level of ovarian fossa) [25]. The grafting site may likely influence the efficacy of the procedure. Most pregnancies (more than 120) were reported after orthotropic grafting [26], which is by far more frequently adopted than heterotopic, and is preferred because it allows also spontaneous conception. After thawed ovarian tissue grafting, the transplanted tissue undergoes neovascularization due to the local production of proangiogenetic factors; as soon as the blood supply takes place, folliculogenesis begins inside the transplanted fragments; estrogenic levels begin slowly to rise, with a consequent decrease in circulating FSH. This process takes about 10–15 weeks to be clinically detectable, and after this time menstruation may restore. This happens in a fairly high proportion of cases, ranging from 67.3% to 93.7% [27,28,29]. Despite the high rate of follicular loss due to post-grafting ischemia, the number of primordial and primary follicles in the grafted tissue is so high that those who survive to ischemic death are enough to guarantee a remarkable endocrine activity for a few years [30,31]. The earliest case of ovarian restoration and live birth using cryopreserved human ovarian fragments was reported in 2000 [32] and, to date, more than 200 births have been reported after the autotransplantation of frozen ovarian tissue [33]. A recent meta-analysis of three centers has calculated a pregnancy rate of about 50% per patient [34] and a live birth rate of 25% [13]. Although a vast majority of women receiving autotransplantation (95%) experience the return of endocrine function, the average duration of ovarian endocrine function is approximately 2–5 years, being related to the OR at the time of OT cryostorage [35]. To increase the number of mature oocytes promptly originated after transplantation, the in vitro activation of dormant follicles (IVA) has been recently proposed [36]. The procedure includes two steps after tissue thawing: the first is based on the complete fragmentation of cortical biopsies in order to promote follicle growth and, at the same time, the progression from secondary to early antral stage by disrupting the Hippo signaling pathway [37]. The second step involves the in vitro culture of cortical pieces with a mix of both PTEN inhibitor and PI3K activator, in order to stimulate the activation of dormant primordial follicles [38]. This procedure has been reported by Suzuki et al., who successfully restored fertility in patients diagnosed with POI after auto-grafting of vitrified human ovarian tissue coupled with follicles obtained by the IVA procedure [39]. Interestingly, this fascinating strategy would fit mainly to women undergoing OTC and reimplantation in more advanced fertile age.

Although the main scientific societies (American Society for Reproductive Medicine and European Society of Human Reproduction and Embryology) have established that OTC has completed the experimental stage and should now be referred to as usual care [14,40], OTC still faces many challenges in clinical application, especially for cancer patients [41]. The frozen ovarian tissue, which is generally stored before the start of cancer treatment or after the remission induction phase, carries the risk of harboring metastasized malignant cells [42,43,44]. After thawing and subsequent transplantation of the OT, these micrometastases have shown the capacity to develop into recurrent tumors in the mouse model, indicating that the autotransplantation of frozen-thawed ovarian tissue could lead to the reintroduction of the malignancy [45]. With the rising number of OTT, the urgency to develop strategies directed at the elimination of malignant cells or to offer alternatives to autotransplantation has become even more evident [46]. A safe fertility restoration program based on OT cryopreservation and transplantation should necessarily take into account the contamination of malignant cells potentially leading to cancer recurrence after allografting. 

This review describes brand new strategies that could be applied in all clinical situations, including cancers metastasizing to the ovary: construction of an artificial ovary (AO) and in vitro maturation of isolated ovarian stem cells (OSCs). 

## 3. Artificial Ovary

Follicle-containing scaffolds are referred to as an “artificial ovary” (AO) and represent a hope for all young female cancer patients, in particular prepubertal girls with a disease able to metastasize to the ovary [47]. Indeed, the transplantation of isolated follicles enclosed in a suitable scaffold rather than of the entire OT may represent an intriguing strategy to eliminate the risk of malignant cell reintroduction [48]. The construction of an AO aims to restore the ovarian three-dimensional structure, supporting follicle survival and development, ensuring secretion of sex hormones and leading to the production of fertilizable, mature oocytes [49]. Briefly, AO technology is based on the isolation of follicles from the ovarian cortex (fresh or thawed after cryostorage) and on their subsequent embedment in an artificial matrix, with the aim of preserving their viability and, once the AO has been grafted to the patient, ensuring functional activity (gametogenesis and hormonal secretion) (Figure 1). Notably, reagents employed to isolate ovarian follicles must fully comply with good manufacturing practice (GMP) guidelines in order to allow application in the clinical practice. The optimal procedure to maximize the efficiency of the isolation process of intact preantral follicles from the surrounding ovarian stroma is still a matter of debate: only a few studies reported the final follicular recovery rate, and the efficiency of different candidate biomaterials is still under evaluation [50]. To date, the best method is considered a combination of mechanical and enzymatic tissue digestion [51]: a mixture of the enzyme liberase and tumor dissociation enzyme (TDE) was reported to be able to yield a high number of viable follicles, with a limited proportion of cells showing apoptosis or necrosis [52,53]. As liberase-isolated follicles tend to clump together, entrapping malignant cells in the aggregate, a washing step performed immediately after TDE treatment is necessary to reduce the risk of malignant cell contamination [54]. The ovary contains follicles at different developmental stages, ranging from primordial to antral; secondary follicles are superior to primordial/primary follicles in terms of survival and post-grafting growth potential. Unfortunately, secondary follicles are very scarce in comparison with primordial/primary, especially in frozen/thawed OT; a scaffold matrix must include enough follicles to produce mature oocytes after transplantation, but not too many, in order to maintain a small-sized fragment that will be grafted and allow proper vascularization of all follicles by a vascular network that is forcedly underdeveloped. Vascular development (neoangiogenesis), in fact, is essential to prevent ischemic injury to the follicles after grafting [55]. Isolated ovarian follicles can survive and develop if nourished by newly formed vessels and supported by bidirectional cross-talk with surrounding stromal cells and endothelial cells, the former providing a suitable paracrine environment for the graft, the latter improving neovascularization [56]. Indeed, ovarian stromal cells are known to differentiate into theca cells, important for androgen hormone production, the substrate of aromatization, and finally for follicular development [57]. Neovascularization in the grafted AO can be boosted in small-sized matrix fragments including endothelial cells, which can be easily obtained from the medullary part of the ovary [58]. Therefore, combining stromal and endothelial cells is of pivotal importance to increase the efficiency of the transplanted AO [59]. 

The choice of the 3D matrix of an AO is probably the most challenging point. Ideally, the scaffold of AO must be composed either of synthetic or natural polymers and must provide adequate protection and support to the included follicles, meeting bio-safety and adaptability to human body temperature. It must also be fully tolerated by the host immune system, minimally degradable after implantation, and statically able to sustain follicle growth and promote neovascularization by providing oxygen and a nutrient supply to follicular, stromal, and endothelial cells [48,49,60,61]. Some synthetic polymers have the possibility of adapting their mechanical properties to the specific requirements of this clinical application, but do not contain molecules essential for cell adhesion, which must be added afterwards [60,61]. To date, the only synthetic polymer that has been utilized to graft isolated preantral follicles is ethylene glycol (PEG) [62]. For a comprehensive description of the available matrixes for AO construction see the recently published review by Dadashzadeh et al. [63]. In comparison to synthetic polymers, natural polymers show a better interaction with cells, thanks to the presence of bio-functional molecules promoting cell adhesion, migration, proliferation, and differentiation. On the other hand, natural matrixes have some disadvantages, such as the reduced mechanical strength and the lower adaptability linked to their complex structure [61]. The first natural matrix ever used to graft isolated preantral follicles was collagen [64]; later, other natural matrixes, such as plasma clots and fibrin, were tested [65,66,67] in combination with collagen or alginate [68,69] loaded with vascular endothelial growth factor (VEGF) [70], platelet lysate [71] or fibrinogen [67]. Also, decellularized OT has yielded successful outcomes with isolated mouse follicle transplantation [72,73,74]; recently, Pors [75] reported promising results with human preantral follicles seeded inside decellularized human OT; isolated follicles were able to survive three weeks after xenografting to mice. The main problem of decellularized OT matrix is the variable size of preantral follicles (30–150 µm), making them hardly adaptable to the pores of the matrix. One alternative is to transform this matrix into a thermosensitive hydrogel, an approach that allows perfect encapsulation of isolated follicles, while retaining the composition of the matrix: in vitro experiments using this procedure demonstrated that isolated mouse preantral follicles could successfully survive in decellularized OT [76,77]. Some studies demonstrated increased estradiol (E2) or progesterone (P4) compared to fresh ovaries in vitro [78,79] whereas others observed that follicular development and ovarian function of decellularized-based artificial ovaries contributed to higher E2 or P4 or inhibin A compared with those in ovariectomized mice, also in vivo [72,74,80].

The cryopreservation of AOs before grafting may be accomplished using an acceptably efficient protocol [81]. Fresh or thawed AOs may be grafted at different transplantation sites, either orthotopic (e.g., pelvic cavity, ovary, or peritoneal window at the ovarian level), or heterotopic (e.g., forearm, neck, or abdominal rectum muscle) [82]. The two major concerns after transplantation are immune rejection and ischemic injury. As transplanted ovarian cells in the AO are of an autologous origin, their graft should be well accepted, but the immune reaction against the matrix is still a concern [83]. As for the ischemic damage after grafting, the recipient’s treatment with VEGF, Vitamin E and gonadotropins resulted in a better transplantation outcome [84]. More recently, human OT was co-transplanted with engineered endothelial cells able to constitutively express anti-mullerian hormone (AMH) to limit the metabolic activation of primordial follicles; these cells were also able to promote the neo-formation of vessels at the host/graft interface [85]. This technology represents a successful cell-based strategy that combines accelerated perfusion of the grafted tissue with paracrine regulation. Another study in SCID mice reported the use of macroporous alginate scaffolds supplemented with bone morphogenetic protein-4 (BMP-4); follicles that developed after grafting reached the antral size and secreted estrogens leading to the restoration of ovarian function [86]. Although procedures have been optimized in the last decades, the success rate of thawed OT transplantation is still limited [87]. The graft’s functional lifespan depends on several factors: age and OR at the time of cryopreservation, OT freeze-thawing protocol, follicle survival rate after thawing, amount of grafted OT, transplantation site and technique, and degree of ischemia and apoptosis after grafting [88]. As a matter of fact, post-graft OT survival is relatively short (2–3 years on the average). Although a rate of nearly 33% of giving live birth was observed in female mice [89], case reports of live births in patients requiring artificial ovaries are still lacking to support the safety of this strategy. However, the use of scaffold as a carrier to preserve the developmental potential of primordial germ cells may represent a potentially effective method for preserving fertility in prepubertal girls or as a model to further investigate the molecular mechanisms induced by gonadotoxic treatments [90].

## 4. In Vitro Maturation of Isolated Ovarian Stem Cells (OSCs)

Experimental research on animal models suggested an active role for stem cells in cases of human OT grafting, leading to the hypothesis that they could improve the autograft outcome. Among all stem cells available in the body, adipose-derived mesenchymal stem cells (ASCs) are known to be easily obtainable and are able to differentiate into multiple lineages [91]. Recent studies investigated the loading of ASCs inside a fibrin scaffold to support human OT transplantation in nude mice; this support was shown to operate as a substrate to prepare a peritoneal grafting site by enhancing oxygenation and vascularization, and to be effective in decreasing post-transplantation ischemic injury and apoptosis, enhancing primordial follicle survival up to 62% one week after transplantation [92,93]. Interestingly, ASCs promoted neovascularization in grafted OT by increasing VEGF and FGF2 and, possessing the functional phenotype of mesenchymal stem cells (MSCs), were able to differentiate into different cell lineages, included endothelial cells [94]. Enhanced angiogenesis contributed to increase follicle survival rate, while decreasing apoptosis and follicular activation, finally better preserving the primordial follicle pool compared to currently used transplantation procedures. These results showed that the lifespan and quality of OT grafts can be improved, potentially extending patients’ fertility and ovarian endocrine activity for longer periods post-transplantation. 

Overall, there are three ways to utilize MSCs for ovarian tissue transplantation [95]: (i) OT is implanted on the peritoneum, then MSCs are injected directly into the center of the transplanted OT fragment; (ii) a fibrin scaffold is prepared with MSCs and implanted on the inner peritoneal surface; 14 days later, OT is placed between the scaffold and the peritoneum; and (iii) OT is implanted into the subcutaneous area of the abdomen, MSCs are placed under the graft with growth factor-reduced Matrigel. All types of MSCs may help neovascularization and blood perfusion of transplanted grafts; however, direct injection of those derived from the adipose tissue (ASCs) into the OT may increase apoptosis, possibly due to overstimulated inflammatory response [96]. Noticeably, bone marrow-derived mesenchymal stem cells (BM-MSCs) transplanted with human OT were shown to stimulate neovascularization in a softer way, via enhanced expression of VEGF, FGF2 and angiogenin; they finally supported functional blood perfusion within the transplanted tissue, decreasing apoptosis in primordial follicles [97]. Indeed BM-MSCs established a persistent and long-lasting pro-angiogenic microenvironment [98,99]. Besides the effect on tissue neovascularization, MSCs were also shown to directly stimulate neo-folliculogenesis: umbilical cord mesenchymal stem cells (UC-MSCs) embedded into a collagen matrix and then transplanted into mice with POF, were able to significantly increase circulating ovarian steroid levels and increased the follicle number, inducing granulosa cell proliferation and ovarian neo-angiogenesis [100]. Also, MSCs isolated from the mouse skin were observed to differentiate into ovarian cell-like cells, and that after being transplanted to ovariectomized mice formed new follicles and blood vessels within eight weeks, restoring the hormonal activity and the estrus [101].

Another interesting issue is the possible participation of resident, oogonial stem cells (OSCs), to the functional resumption of ovarian activity. OSCs, also known as germline stem cells, were identified inside the surface epithelium of mouse ovaries, where they constitute only a very small percentage of all cells (0.014%) and become even scarcer with increasing age [102]. The paradigm that the number of oocytes is fixed before birth and cannot increase throughout life [103] was questioned when OSCs were isolated and characterized also in the human ovary, leading to the hypothesis of the existence of novel oocyte production after birth [104,105]. A growing body of evidence supports the idea that the reintroduction of OSCs back into OT following chemotherapy may enable at least a partial recovery of normal ovarian function. The first observation made to this regard came from mouse studies, in which OSCs were isolated from adult ovaries and submitted to long-term in vitro propagation, with the final generation of fertilization-competent eggs following intraovarian transplantation [105,106]. White et al. injected mouse GFP-expressing OSCs into OT fragments that were then xenotransplanted into NOD–SCID mice previously treated with chemotherapy; within 1–2 weeks the authors observed their differentiation into follicles containing GFP-positive oocytes, which completed maturation and produced viable offspring after in vitro fertilization [107]. The hypothesis that OSCs isolated from the ovary can differentiate in vitro into newly formed oocytes suggests an innovative therapeutic application for female infertility [108]. Interestingly, studies performed on murine models suggested that isolated OSCs can be fertilized and produce embryos in vitro [109,110,111]. 

However, the identification and validation of deliverable “oogenic” factors capable of driving OSC differentiation into oocytes is still under investigation. Unfortunately, the list of these factors is quite small at present and includes histone deacetylase inhibitors [112], bone morphogenetic protein 4 (BMP4) [113], and Hippo signalling pathway components [114]. Nevertheless, the ability of OSCs to generate in vitro-derived oocytes in mammalian models provides the motivation to perform high-throughput screening of candidate oogenic factors that can then be rigorously tested for their ability to expand the ovarian reserve in vivo [115,116]. In humans, under appropriate in vitro culture conditions Ddx4-positive OSCs from non-menopausal and menopausal women were observed to differentiate into large haploid oocyte-like cells expressing the major oocyte markers growth differentiation factor 9 (GDF-9) and synaptonemal complex protein 3 (SYCP3): these oocyte-like cells were also able to enter meiosis [117,118]. Interestingly enough, the Ddx4-positive cell population was detected in comparable proportion in fresh and vitrified/thawed cortical OT, maintaining excellent post-thaw viability and the potential capability of generating mature oocytes in vitro [119]. In a different set of experiments, follicle-stimulating hormone (FSH) and basic fibroblast growth factor (bFGF) were shown to promote the transition of primordial follicles into primary follicles in cultured human OT; OSCs retained the potential to spontaneously differentiate into oocyte-like structures in extended cultures [120]. To the best of our knowledge, the first report of the culture of OSCs in a transplanted decellularized ovarian matrix was recently provided by Mirzaeian et al.; these authors showed the development of oocyte, granulosa and endothelial cells forming follicle-like structures after OSCs culture with or without addition of peritoneal and bone marrow mesenchymal stem (PMSCs and BMSCs, respectively) [121]. The above-mentioned evidence suggests that the implantation of OSC-derived oocytes into the ovaries of young women with POI could represent a promising strategy to restart follicle regeneration and consequently recover ovulation and fertility [122,123]. Interestingly, different IVM protocols have been studied next to the conventional COCs collection by transvaginal follicle aspiration such as in situ follicle aspiration from the ovary or the contralateral ovary during laparoscopy/laparotomy and ex vivo aspiration either from the excised ovarian tissue or the spent media during cortex preparation before cryopreservation. The differential benefits between the monophasic (OTO-IVM) and biphasic (CAPA)-IVM) protocols have been recently reviewed [124]. However, concern still exists about the possibility that obtaining novel oocytes in vitro might interfere with the complex genomic imprinting and epigenetic mechanisms required for the development of fully competent oocytes [125]. Although fascinating, the idea of ovarian rejuvenation by the injection of in vitro-differentiated OSCs is still intensely debated, and the relatively scarce evidence available so far outlines the need of further experimental work before hypothesizing a clinical application [126,127]. 

## 5. Future Challenges 

Besides FP, the AO construction and the OSC isolation from the ovarian cortex could also offer future conception chances to women developing POI due to genetic predisposition or benign ovarian diseases, or forced to undergo repeated ovarian surgery [128,129,130,131,132]. However, several limitations still have to be overcome in this field: (i) technical: the choice of a suitable matrix for an AO and the selection of specific markers for OSC isolation are the most challenging topics; (ii) clinical: many more preclinical research studies are needed before a routine clinical application of AO and OSCs could become practically available; and (iii) ethical and legal: all the possible risks of these technologies should be evaluated at each step, not only on the patient herself but also on the future offspring as ex vivo manipulations with ovarian follicles might lead to genetic and epigenetic changes in the oocytes, which might directly affect the offspring’s health. Quite often malignant diseases are hereditary, and these technologies would contribute to the transmission of genetic variants associated with cancer to the next generation and therefore would potentially increase the percentage of patients with cancer. Moreover, a patient might refuse AO transplantation or injection of OSC-derived oocytes for personal or financial reasons, but the use of such expensive technologies might emotionally pressure and oblige a patient to motherhood. On the other hand, both ethical and legal questions may arise in case of a patient’s death, or in cases of demands of donation for scientific research or pharmaceutical drug testing. Also, their commercialization might be associated with moral issues as the problem of donor organ shortage would be solved, but an issue of the accessibility to the entire population would arise. Finally, the definition of inclusion criteria to access these strategies would raise a number of moral questions. 

## 6. Conclusions

In conclusion, these strategies represent potentially innovative and promising strategies to preserve gonadal function from anticancer treatment negative effects [133,134]. Both technologies could be used to avoid the risk of re-implanting malignant cells in OT allograft as well as the ovarian hormonal stimulation in case of estrogen-sensitive tumors [135,136]. 

## Figures and Tables

**Figure 1 healthcare-11-02748-f001:**
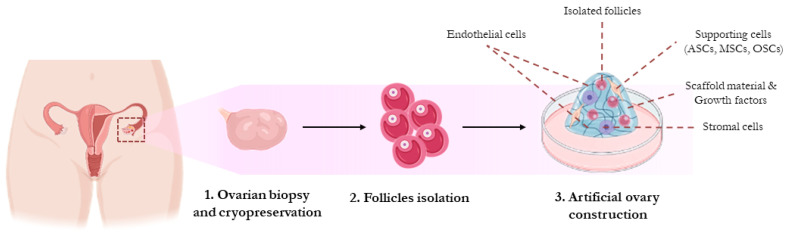
Artificial ovary construction. The artificial ovary aims to restore reproductive and/or endocrine function of ovaries. Ovarian tissue or entire ovaries are retrieved and cryopreserved. Isolated follicles are obtained and cultured in a three-dimensional scaffold (the so-called artificial ovary) that permits follicles survival and development. To increase artificial ovary efficiency, growth factors, stromal and endothelial cells as well as supporting cells can be also included into the scaffold to permit hormone production and enhance vascularization.

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
