# Peer review of "Innovative Strategies for Fertility Preservation in Female Cancer Survivors: New Hope from Artificial Ovary Construction and Stem Cell-Derived Neo-Folliculogenesis"

_healthcare, 2023, doi:10.3390/healthcare11202748_

Round 1

Reviewer 1 Report

The topic of the review manuscript is contemporary and treats on important issue in reproductive medicine. Therefore, I think that the paper will be useful to the scientists working in the field. The manuscript is well structured and written in sound scientific style. The most contemporary achievements in the field have been adequately described. 

However, I have the following remarks:

1. In the section about ovarian tissue cryopreservation, the authors give preference to the vitrification procedure as more beneficial for cryopreservation of ovarian tissue, compared to slow freezing. Only papers, which confirm that point of view have been cited, but many others have been published (including new ones), which point out to the opposite conclusions. I would like to state, that the worldwide practice is predominantly with programme (slow) freezing of ovarian tissue and only 4 children have been born after vitrification. However, I think that the remark is not influencing the merit of the manuscript, as ovarian cryopreservation techniques are not its main focus and I suggest that it should be accepted for publication.

2. There are some typing mistakes, e.g. in line 89 instead of "low freezing" should be "slow freezing".

Author Response

The topic of the review manuscript is contemporary and treats on important issue in reproductive medicine. Therefore, I think that the paper will be useful to the scientists working in the field. The manuscript is well structured and written in sound scientific style. The most contemporary achievements in the field have been adequately described. 

We sincerely thank the reviewer for the consideration.

However, I have the following remarks:

1. In the section about ovarian tissue cryopreservation, the authors give preference to the vitrification procedure as more beneficial for cryopreservation of ovarian tissue, compared to slow freezing. Only papers, which confirm that point of view have been cited, but many others have been published (including new ones), which point out to the opposite conclusions. I would like to state, that the worldwide practice is predominantly with programme (slow) freezing of ovarian tissue and only 4 children have been born after vitrification. However, I think that the remark is not influencing the merit of the manuscript, as ovarian cryopreservation techniques are not its main focus and I suggest that it should be accepted for publication.

We agree with the reviewer’s valuable comments. We rephrased some sentences in that paragraph to provide a clearer message of caution.

2. There are some typing mistakes, e.g. in line 89 instead of "low freezing" should be "slow freezing".

We apologize for this. We have corrected the typing mistake and revised the manuscripts for others.

Reviewer 2 Report

This review investigates some interesting techniques to keep fertility in female young cancer patients. The authors discussed both advantages and disadvantages of 3 main techniques: ovarian stem cell isolation method, ovarian cell cryopreservations and artificial ovarian method.

Major issues:

1, it will be better if more research about comparisons between artificial ovarian cells and biological ones are involved to be discussed or presented in the manuscript.

2, Authors should also include more clinical applications of those techniques in the manuscript if possible.

3, In the fisrt section of the reivew, authors should discuss something about current scenarios about preservation of fertility under present clinical settings.

Minor issues:

1. there are some word spelling errors in abstract section

2. In the last part, you should divide the paragraph into future challenges and conclusions instead of combining them together.

Author Response

This review investigates some interesting techniques to keep fertility in female young cancer patients. The authors discussed both advantages and disadvantages of 3 main techniques: ovarian stem cell isolation method, ovarian cell cryopreservations and artificial ovarian method.

Major issues:

1, it will be better if more research about comparisons between artificial ovarian cells and biological ones are involved to be discussed or presented in the manuscript.

We thank the reviewer for the suggestion. We have now discussed more reaserch on the comparison between artificial ovarian cells and biological ones.

2, Authors should also include more clinical applications of those techniques in the manuscript if possible.

We thank the reviewer for the suggestion. We have now provided further clinical application of AO

3, In the fisrt section of the reivew, authors should discuss something about current scenarios about preservation of fertility under present clinical settings.

We thank the reviewer for the suggestion. We have now implemented the first paragraph with the current approaches available for fertility preservation.

 Minor issues:

  1. there are some word spelling errors in abstract section

We apologize for this. We have corrected the typing mistakes and revised the manuscripts for others.

  1. In the last part, you should divide the paragraph into future challenges and conclusions instead of combining them together.

We thank the reviewer for the suggestion. We have now divided the two paragraphs as requested.

Reviewer 3 Report

I think this report is meaningful, because infertility treatments are important for the current trend of later marriage and delivery. So, some other review articles can be found and I have some requests as follows.

1: About artificial ovary, I want to know the possibility of giving live birth, if possible, about human. In the other report which introduce artificial ovary, they wrote “Nearly 33% of female mice deliver offspring”. (Jing Chen et al. Artificial Ovary for Young Female Breast Cancer Patients)

2: I think in vitro maturation of immature oocytes (IVM) show the sufficient therapeutic effect for patients who cannot give birth by IVF. And the difference between OSC and IVM and the outcome of treatment are described in the article (Chloë De Roo et al. In Vitro Maturation of Oocytes Retrieved from Ovarian Tissue: Outcomes from Current Approaches and Future Perspectives).

Author Response

I think this report is meaningful, because infertility treatments are important for the current trend of later marriage and delivery. So, some other review articles can be found and I have some requests as follows.

 1: About artificial ovary, I want to know the possibility of giving live birth, if possible, about human. In the other report which introduce artificial ovary, they wrote “Nearly 33% of female mice deliver offspring”. (Jing Chen et al. Artificial Ovary for Young Female Breast Cancer Patients)

We thank the reviewer for the valuable comment. However, to the best of our knowledge no case reports of live births in patients recurring to artificial ovary have been published so far.

 2: I think in vitro maturation of immature oocytes (IVM) show the sufficient therapeutic effect for patients who cannot give birth by IVF. And the difference between OSC and IVM and the outcome of treatment are described in the article (Chloë De Roo et al. In Vitro Maturation of Oocytes Retrieved from Ovarian Tissue: Outcomes from Current Approaches and Future Perspectives).

We thank the reviewer for the suggestion. We have improved the discussion of OSCs maturation approaches including the cited reference.

Reviewer 4 Report

Dear Author, 

I have read with great interest the manuscript titled "Innovative Strategies for Fertility Preservation in Female Cancer Survivors: New Hope From Artificial Ovary Construction and Stem Cells-Derived Neo-Folliculogenesis." This is a literature review on a highly interesting and current topic. The manuscript explores the latest advancements in two technologies aimed at restoring ovarian function following iatrogenic damage (chemotherapy, radiotherapy) in oncological patients: Artificial ovary and In vitro maturation of isolated ovarian stem cells (OSCs). 

The review is clear and well-written, and the topic is both interesting and innovative. The data presented are up-to-date and reflect the most recent scientific discoveries. 

I have only a few comments to make on this review: 

- Regarding paragraph 2, mention other fertility preservation techniques: ovarian transposition and in vitro maturation of cryopreserved oocytes. Please clarify 

- Include epidemiological data on the prevalence of reproductive age women affected by cancer and women who resort to fertility preservation techniques (oocyte cryopreservation and ovarian tissue cryopreservation).Please add these informations. 

- Regarding the artificial ovary, are there in vitro or animal models studies demonstrating follicular growth and oocyte maturation? Please clarify 

- It has been demonstrated that in murine models, OSCs can be fertilized and produce embryos, please add the citations (Stimpfel et al., 2013; Parte et al., 2014; Hernandez et al., 2015) 

- What are the main limitations (technological, clinical, ethical, regulatory, etc.) of these innovative procedures?

Author Response

I have read with great interest the manuscript titled "Innovative Strategies for Fertility Preservation in Female Cancer Survivors: New Hope From Artificial Ovary Construction and Stem Cells-Derived Neo-Folliculogenesis." This is a literature review on a highly interesting and current topic. The manuscript explores the latest advancements in two technologies aimed at restoring ovarian function following iatrogenic damage (chemotherapy, radiotherapy) in oncological patients: Artificial ovary and In vitro maturation of isolated ovarian stem cells (OSCs). 

The review is clear and well-written, and the topic is both interesting and innovative. The data presented are up-to-date and reflect the most recent scientific discoveries. 

We sincerely thank the reviewer for having appreciated our manuscript.

I have only a few comments to make on this review: 

- Regarding paragraph 2, mention other fertility preservation techniques: ovarian transposition and in vitro maturation of cryopreserved oocytes. Please clarify 

We thank the reviewer for the suggestion. We have now implemented the first paragraph with the current approaches available for fertility preservation.

- Include epidemiological data on the prevalence of reproductive age women affected by cancer and women who resort to fertility preservation techniques (oocyte cryopreservation and ovarian tissue cryopreservation).Please add these informations. 

We thank the reviewer for the suggestion. We have now added the requested informations in the text.

- Regarding the artificial ovary, are there in vitro or animal models studies demonstrating follicular growth and oocyte maturation? Please clarify 

We thank the reviewer for the comments. We have now added some studies suggesting follicula growth and oocyte maturaton using artificial ovaries compared to fresh ones.

- It has been demonstrated that in murine models, OSCs can be fertilized and produce embryos, please add the citations (Stimpfel et al., 2013; Parte et al., 2014; Hernandez et al., 2015) 

We thank the reviewer for the suggestion. We have now added the cited references.

- What are the main limitations (technological, clinical, ethical, regulatory, etc.) of these innovative procedures?

We thank the reviewer for the suggestion. We have now added the militations of these strategies in the “future challenges” chapter

Round 2

Reviewer 3 Report

Thank you for revising the article. I have no other request.